# Constrained Pulse Radar Waveform Design Based on Optimization Theory

**DOI:** 10.3390/s25041203

**Published:** 2025-02-16

**Authors:** Jianwei Wu, Jiawei Zhang, Yifan Chen

**Affiliations:** 1School of Information Science and Engineering, Yanshan University, Qinhuangdao 066004, China; wujianwei@stumail.ysu.edu.cn (J.W.); zhangjw@ysu.edu.cn (J.Z.); 2School of Life Science and Technology, University of Electronic Science and Technology of China, Chengdu 610054, China

**Keywords:** transmit waveform, constrained optimization, scattering model, criterion, optimization algorithm

## Abstract

Radar is utilized as an active sensing device across many fields. Its waveform optimization is responsible for target signature extraction, profoundly influencing the overall performance. First, the principle of pulse radar waveform design is explored. Waveform design strategies vary based on target models, whether point-like or extended ones, and are often formulated as high-dimensional, non-convex optimization problems with multiple constraints, such as energy, constant modulus, and sidelobe ratios. Second, to address them, techniques like alternating direction method of multipliers (ADMM), semidefinite relaxation (SDR), and minimization-maximization (MM) algorithms are widely employed. Finally, challenges in multimodal sensing collaborative detection, joint multi-tasking, sparse signal recovery, and intelligent perception highlight the need for innovative solutions to meet future demands.

## 1. Introduction

As an active sensor, radar interrogates the environment by transmitting a customized waveform and processing echo data to determine the presence of target, location, and other signatures. Nowadays, radar is widely utilized in various fields, including security-related tasks, planetary exploration, geoscience research, climate monitoring, traffic surveillance, severe weather detection, earth resources management, automotive safety, etc. [1].

A general radar system includes a transmitter, antenna, receiver, and signal processor as shown in Figure 1. It operates by emitting waveforms and processing the echoes that bounce back from targets. A good transmitted waveform can not only increase the precision of target recognition but also reduce receiver complexity [2].

Radar performance enhancement through waveform optimization has been a focal point of research since 1965, and it has been demonstrated that an appropriately designed transmitted waveform is more important than receiver optimization. Since then, there has been a sustained interest in designing pulse radar transmit signals aiming at enhancing target signatures [3]. In essence, waveform optimization involves redistributing emission energy to allocate less energy to regions with strong interference, noise, or clutter, while concentrating more energy on areas of interest, achieving the detection of relevant scatterers as well as suppressing clutter [4]. Therefore, the waveform design issue is always boiled down to an optimization problem, where performance metrics always serve as the objective function, subject to some reasonable constraints.

The research in pulse radar waveform optimization has been quite fertile from the perspectives of different target scattering models, optimization criteria, and solving algorithms.

The models of radar operation include point-like and extended targets, based on the relationship between the spatial resolution of the radar and the physical size of the target. Under the point target assumption, physical characteristics and geometry of the target are ignored. The transmitting waveforms widely used in point targets always desire a wideband property, and the main goal is to obtain a high resolution and a low sidelobe output. In contrast, the echo of the extended target is no longer simply the scaled transformation of the transmitted waveform, but the convolution of the transmitted waveform and the target impulse response (TIR). Waveforms designs under such a model can be grouped into two categories including the deterministic TIR and the stochastic TIR.

Depending on the tasks of radar, optimization criteria for waveform design are mainly divided into three categories: signal power, information theory and ambiguity function, where the constraints may also present multiple ones, including energy, constant modulus, low peak-to-average-power ratio (PAPR), spectrum, and similarity constraints. They are usually formulated as a multi-constraint non-convex high-dimensional nonlinear optimization problem, making it difficult to find the optimal solution within polynomial time complexity. To tackle the resultant issue, some customized methods are developed, by resorting to alternating direction method of multipliers (ADMM), semidefinite relaxation (SDR) and minimization-maximization (MM).

Although it is unrealistic to find a single radar waveform design method that fits all types of radar systems, the waveform architectures and optimization methods designed for pulse radar are generally applicable to improve system performance. For example, in target detection, it is necessary to improve the signal-to-noise ratio (SNR) and signal-to-clutter ratio (SCR), which can enhance the distinction between target scattering and interference; for tracking radar, the designed waveform should be capable of rapidly updating, as there are high demands for real-time waveform changes; in the case of imaging radar, a large bandwidth is always required to achieve high-resolution imaging and ensure the acquisition of detailed target information.

Radar waveforms are widely applied across multiple fields. In space exploration, large bandwidth unimodular waveforms facilitate high-resolution imaging of celestial bodies and accurate planetary mapping. In climate monitoring, real-time waveform optimization enhances sensitivity to dynamic weather phenomena, aiding disaster warning and management. In geological research, waveform optimization improves scanning resolution and sensitivity, supporting precise mineral exploration. In earth sciences, optimized waveforms are applied in glacier movement research, soil moisture measurement, and terrain mapping.

Despite significant progress, radar waveform design still faces challenges. The rise of multimodal sensing technologies, such as integrating radar with optical sensors and lidar, has introduced the need for coordinated waveform optimization for collaborative detection. Modern radar applications also demand multitasking capabilities, requiring waveforms to support diverse functions like detection, imaging, and tracking. Sparse signal recovery and waveform adaptation in dynamic environments remain critical issues. Due to limitations in real-time performance and computational complexity, radar waveform optimization for intelligent environmental perception faces significant challenges. Future research may focus on multi-objective optimization and the development of adaptive algorithms to enhance performance in complex and dynamic environments.

Current breakthroughs are reflected in several areas. Research in waveform design has progressed from deterministic approaches to incorporating non-deterministic prior information, enabling greater adaptability to real-world complexities. Advanced optimization algorithms and the transition from single-objective to multi-objective optimization have significantly enhanced system performance. Additionally, growing focus on electronic interference has refined waveform design to address these challenges effectively, driving advancements in both theory and application.

In this paper, a comprehensive study on radar waveform design is presented, highlighting the significance of waveform optimization for enhancing radar system performance. We firstly explore the fundamental principles of radar waveform design, with a particular focus on the design requirements for point and extended targets. For point targets, the primary objectives of waveform design are to enhance resolution and minimize sidelobes, thereby optimizing target detection and identification. In contrast, waveform design for extended targets is influenced by the TIR, requiring consideration of the target’s physical characteristics and geometry. Secondly, the paper systematically formulates radar waveform optimization as a high-dimensional, non-convex problem, incorporating various constraints such as energy, constant modulus, and sidelobe ratios. To address them, a critical review of several advanced optimization techniques is provided, including the ADMM, SDR, and MM algorithms, detailing their applications in radar waveform optimization. Among them, ADMM decomposes the problem to facilitate efficient solutions, SDR relaxes non-convex constraints into convex problems for tractability, and MM iteratively converges toward the optimal solution. Finally, we discuss emerging challenges in radar systems, such as multimodal sensing, collaborative detection, joint multitasking, sparse signal recovery, and intelligent perception, and present potential solutions to address these issues.

The rest of this paper is organized as follows. Section 2 introduces the principle of pulse radar waveform design. In Section 3, some representative radar waveform optimization methods are presented. Some key challenges are discussed in Section 4 and conclusions are drawn in Section 5.

## 2. Principles of Pulse Radar Waveform Design

Radar waveform design can be concepualized as the formulation and solution of an optimization problem, as shown in Figure 2. Based on task requirements, target scattering models and engineering implementation, appropriate optimization criteria and constraints are determined, followed by utilization of optimization algorithms to generate the desired waveform.

### 2.1. Target Scattering Models

A point target can be analogized to a metal sphere. Optical images provide an intuitive representation of the geometric structure and physical appearance of the targets, while inverse synthetic aperture radar (ISAR) image reflects the backscatter coefficient of the point target. The scattering characteristics of an extended target are closely related to its shape, posture, and material. ISAR image illustrates the backscatter coefficient of the extended target under specific frequency and plane wave source. Analyzing these images allows us to evaluate how the radar waveform interacts with the targets, which in turn aids in optimizing the radar waveform design.

#### 2.1.1. Point-like Target

If a point-like target with an infinitesimal physical size is illuminated by radar, the reflected echo is similar to the transmitting signal but carrying on a certain delay and Doppler shift. Assuming that the transmitted signal is $x\left(t\right)=a\left(t\right)e^{j2\pi f_{c}t}$, where a(t) and $f_{c}$ are the envelope and carrier frequency, respectively, the echo of a point-like target is written as(1)$$y\left(t\right)=\sqrt{E_{R}}a\left(t-\tau \right)e^{j2\pi (f_{c}+f_{d})t},$$
where $E_{R}$ represents energy, τ and $f_{d}$ are the time delay and Doppler shift, respectively. The scattering characteristics of a point-like target can be approximately described by an isotropic metal sphere, where an ISAR image of an ideal metal sphere is provided in Figure 3.

#### 2.1.2. Extended Target

When the size of a target exceeds the radar resolution, the echo will not be a monochromatic wave, but have different frequency response components. Then, the point target model no longer holds and such targets are termed extended targets [5]. The TIR h(t,θ,φ) of an extended target is dependent on the target attitude, and is therefore a function of azimuth θ and pitch φ. When the transmitting signal is x(t), the echo of an extended target is(2)$$y\left(t\right)=x\left(t\right)*h\left(t\right)=\int_{-\infty }^{\infty }h\left(\tau \right)x\left(t-\tau \right)d\tau$$
Referring to Figure 3, we also provide an ISAR image of an extended target in Figure 4.

### 2.2. Optimization Criteria

By choosing appropriate criteria, detection performance, resolution capability, and anti-jamming ability of the waveform can be improved, enhancing the overall performance of the radar system. Thus, a wide range of optimization criteria are involved in waveform design, which can be divided into signal power, information theory and ambiguity function.

**Signal-Power-Based Criteria:** This group mainly includes SNR and signal-to-interference-plus-noise ratio (SINR). Since a higher SNR or SINR leads to a better detection probability, these metrics are widely used to benefit target detection. Considering background clutter c(t) and noise n(t), the generalized echo signal model can be represented by(3)y(t)=x(t)∗h(t)+x(t)∗c(t)+n(t)
Obviously, the SINR is defined as(4)$$\mathrm{SINR}=\frac{|x\left(t\right)*h\left(t\right){|}^{2}}{E\left[|x\left(t\right)*c\left(t\right){|}^{2}\right]+E\left[|n\left(t\right){|}^{2}\right]}$$
where E[·] denotes the mathematical expectation.

**Information-Theory-Based Criteria:** The mutual information (MI) and Kullback-Leibler divergence (KLD) are representative in this type of criteria [5,6,7,8,9,10,11,12]. Shannon first introduced MI into the communicaiton system, and later Woodward et al. [13] applied it to the field of radar. MI benefits radar parameter estimation and target identification, which indicates that, the higher MI between received echo and target, the more information about target can be obtained [5]. Considering the jointly distributed discrete random variables *X* and *Y*, the MI between *X* and *Y*, denoted I(X;Y), is defined as(5)I(X;Y)=H(X)−H(X|Y)
where *X* and *Y* represent the target and received echo, respectively. H(X) and H(X|Y) are the information entropy of *X* and conditional entropy, respectively. The MI represents the difference between the prior uncertainty in *X* and uncertainty in *X* after observing *Y*, which is regarded as a measure of the amount of information provided about *X*.

Another metric based on information theory is the relative entropy or KLD, which is usually applied in radar target classification. $D(P_{0}||P_{1})$ denotes the relative entropy between hypotheses $H_{0}$ and $H_{1}$, and is written as [14](6)$$\max_{x}\;\;D(P_{0}\left|\right|P_{1})=\int P_{0}\left(y\right)\mathrm{lo}g_{2}\left(\frac{P_{0}\left(y\right)}{P_{1}\left(y\right)}\right)dy$$
where $P_{0}\left(y\right)$ and $P_{1}\left(y\right)$ represent the probability density functions of $H_{0}$ and $H_{1}$, respectively.

**Ambiguity-Function-Based Criteria:** As a common metric for radar waveform design, ambiguity function (AF) reflects the range and Doppler resolution, as well as the power of its sidelobe interference. The AF of a conventional single-antenna radar is defined as(7)$$\chi \left(\tau ,\xi \right)=\int_{-\infty }^{\infty }x\left(t\right)x^{*}\left(t+\tau \right)e^{j2\pi \xi t}dt$$
This 2-D function is the range-Doppler response of the matched filtering to the transmitted signal with time delay τ and Doppler shift ξ. The energy distribution of the AF measures how well a radar system identifies target parameters. More precisely, the sharper the peak of AF, the higher the range-Doppler resolution is acquired.

By definition, the cross-correlation function (CCF) is the cross-section of AF at zero Doppler while the peak sidelobe level ratio (PSLR) and integral sidelobe level ratio (ISLR) are often used to reflect the CCF [15]. Among them, PSLR reflects the highest sidelobe level in pulse compression system relative to the mainlobe level, written as(8)$$\mathrm{PSLR}=10\log_{10}\left(\frac{P_{\mathrm{peak}}}{P_{\mathrm{main}}}\right)$$
where $P_{\mathrm{peak}}$ is the highest sidelobe level, and $P_{\mathrm{main}}$ represents the main lobe level. ISLR represents the average level of all sidelobes relative to the mainlobe [16], with(9)$$\mathrm{ISLR}=10\log_{10}\left(\begin{array}{c} \sum_{k=1}^{N}P_{k}/P_{\mathrm{main}} \end{array}\right)$$
$P_{k}$ and $P_{\mathrm{main}}$ correspond to the kth sidelobe level and the main lobe level, respectively, and *N* denotes the number of sidelobes.

### 2.3. Waveform Constraints

Once the optimization criteria are defined, practical waveform constraints are typically incorporated into the design problem considering hardware configuration as well as application scenarios. Energy, constant modulus, low PAPR, spectrum and similarity constraints are usually imposed, and they are summarized below, in Table 1.

Among them, energy constraint is used to restrain the energy of the transmitting waveform from being too large. Given the limitations of waveform generation hardware, constant modulus is frequently applied, with the PAPR constraint being more general than the unimodular case [17]. Furthermore, to minimize interference from radar systems on coexisting telecommunications, incorporating spectrum constraints in waveform design, is of great significance. A similarity constraint is imposed on the transmitted waveform to regulate various characteristics, including range-Doppler resolution, signal modulus variations, and PSLR.

### 2.4. Optimization Algorithms

#### 2.4.1. ADMM-Based Algorithm

As a method to solve the problem of coupling optimization between variables, ADMM algorithm decomposes the objective function of the original problem into a series of solvable subproblems, allowing for the attainment of a local optimal solution [18]. If the original problem is convex, the ADMM can guarantee global convergence.

Due to the flexible splitting framework of ADMM, it becomes an effective approach for solving waveform optimization problems. A specific waveform optimization for multiple-input multiple-output (MIMO) radar is formulated as [19](10)$$\begin{array}{l} \min_{x}\;\frac{x^{H}Ax}{x^{H}Bx}\\ s.t.\left\{\begin{array}{l} \left|x_{i}\right|=1,i=1,2,\cdots ,N_{t}L\\ {∥x-x_{0}∥}^{2}\leq \varepsilon \\ x^{H}D_{p}x\leq \delta_{p},p=1,2,\cdots ,P \end{array}\right. \end{array}$$
where A and B are viewed as correlation matrices of target and background, respectively. x, $x_{0}$, ε, *L* and $D_{p}$ to represent the transmit waveform, the reference signal, similaity parameter, samples, and a semi-positive definite matrix. The constraints are constant modulus, similarity, and quadratic inequality constraints (with the spectral and PAPR constraints taking this form), respectively.

As we can see, a non-convex problem characterized by a fractional objective function and multiple constraints is given by (Equation 10). Referring to the ADMM framework, the first step is to introduce some auxiliary variables such that(11)$$\begin{array}{c} \min_{x}\frac{x^{H}Ax}{r^{H}Br}\\ s.t.\left\{\begin{array}{c} \left|x_{i}\right|=1,i=1,2,\cdots ,N_{t}L\\ r=x\\ w^{2}\leq \varepsilon ,w=x-x_{0}\\ {v_{p}}^{H}v_{p}\leq \delta_{p},\\ v_{p}={D_{p}}^{1/2}x,p=1,2,\cdots ,P \end{array}\right. \end{array}$$

The equality constraint r=x,$w=x-x_{0},v_{p}=D_{p}^{1/2}x$ are brought into the augmented Lagrange function of problem (Equation 11) [18]. Considering the ADMM ideas, the original variable and the dual variable are successively updated by minimizing and augmenting the Lagrange function [18]. The ADMM framework is to convert the complex optimization problem into a series of simple subproblems, and gradually approach the stagnation point of the original problem by solving the subproblems in turn.

#### 2.4.2. SDR-Based Algorithm

The SDR technique has proven to be highly significant and applicable in pulse radar waveform designs. It serves as a computationally efficient approximation method for addressing a variety of challenging optimization problems [20]. Specifically, it can be utilized for many non-convex quadratically constrained quadratic programs (QCQP) in an almost automatic way. A typical QCQP problem is generally written as(12)$$\begin{array}{c} \min_{x\in R^{n}}\;x^{T}Ax\\ s.t.\;x^{T}B_{i}x\geq b_{i},i=1,\cdots ,n \end{array}$$
where $A,B_{1},\cdots ,B_{n}\in S^{n}$ and $b_{1},\cdots ,b_{n}\in R$. A key initial step in understanding SDR is given by(13)$$x^{T}Ax=\mathrm{Tr}\left(x^{T}Ax\right)=\mathrm{Tr}\left(Axx^{T}\right)=\mathrm{Tr}\left(\mathrm{AX}\right),x^{T}B_{i}x=\mathrm{Tr}\left(x^{T}B_{i}x\right)=\mathrm{Tr}\left(B_{i}xx^{T}\right)=\mathrm{Tr}\left(B_{i}X\right),$$

Both the objective and constraint functions are affine in X. The condition $X=xx^{T}$ is equivalent to X⪰0,rank(X)=1, thus (Equation 12) is(14)$$\begin{array}{c} \min_{X\in S^{n}}\mathrm{Tr}\left(\mathrm{AX}\right)\\ s.t.\left\{\begin{array}{c} \mathrm{Tr}\left(B_{i}X\right)\geq b_{i},i=1,\ldots ,n\\ X⪰0,\mathrm{rank}(X)=1 \end{array}\right. \end{array}$$

#### 2.4.3. MM-Based Algorithm

The MM algorithm is used to address challenging optimization problems by transforming the original complex problem into a sequence of simpler ones, ensuring the results are close to the optimal solution of the original problem. Without loss of generality, a representative optimization problem is written as minf(x),s.t.x∈Θ. At the kth step of iteration, we have $x^{\left(k+1\right)}\in \underset{x\in \Theta }{\arg\min}\;\psi (x,x^{\left(k\right)}$, where $\psi (x,x^{\left(k\right)})$ serves as a majorizing function for f(x) at the point $x^{\left(k\right)}$. The convexity of the constraint set Θ determines the type of stationary points to which the MM technique converges. When Θ is convex, the majorizer is expected to meet (Equation 15) and (Equation 16), ensuring convergence toward a stationary point [21](15)ψ(x,y)≥f(x),for∀x,y∈Θ(16)$$\psi \left(y,y\right)=f\left(y\right),\;\mathrm{for}\;\forall y\in \Theta \;\mathrm{and}\;\psi^{\prime }\left(y,y;h\right)=f^{\prime }\left(y;h\right),\forall h\;\mathrm{with}\;y+h\in \Theta$$
where ψ(x,y)iscontinuouson(x,y), and $f^{\prime }\left(y;h\right)$ represents the directional derivative. If Θ is concave, (Equation 16) requires modification to guarantee the convergence:(17)$$\psi^{\prime }\left(y,y;h\right)=f^{\prime }\left(y;h\right),\;\mathrm{for}\;\forall h\in T_{\Theta }\left(y\right)$$
where ψ and *f* belong to the whole real vector space, and $T_{\Theta }\left(y\right)$ represents the Boulingand tangent cone of Θ at y.

One important characteristic of MM algorithm is monotonicity, i.e.,(18)$$f\left(x^{\left(k+1\right)}\right)\leq \psi \left(x^{\left(k+1\right)},x^{\left(k\right)}\right)\leq \psi \left(x^{\left(k\right)},x^{\left(k\right)}\right)=f\left(x^{\left(k\right)}\right)$$
From (Equation 18), although $x^{\left(k+1\right)}$ is not the minimizer of $\psi (x,x^{\left(k\right)})$, the monotonicity can still be guaranteed once it leads to the function $\psi \left(x^{\left(k+1\right)},x^{\left(k\right)}\right)\leq \psi \left(x^{\left(k\right)},x^{\left(k\right)}\right)$.

## 3. Waveform Optimization Methods

### 3.1. Waveform Optimizations with Different Target Scattering Models

#### 3.1.1. Point-like Target

Based on the spectral characteristics, existing radar waveforms of point-like target can be mainly divided into three categories: linear frequency modulation (LFM), stepping frequency (SF) and nonlinear frequency modulation (NLFM). The LFM signal with the rectangular shape of spectrum, a large time-bandwidth product, simple generation, and good Doppler tolerance, achieves desired point-like target detection performance [22]. In [23], an arbitrary correlated set of LFM waveforms was used to transmit beampattern design, which did not guarantee the constant-envelope property. As an extension, a method for synthesizing the transmit beampattern of MIMO radar through LFM waveforms was proposed in [24], requiring only the optimization of pulse duration, frequency steps, and original phases of the LFM. To improve signal to clutter noise ratio in a cognitive MIMO radar system, the LFM waveform and receiver impulse response were jointly optimized in [25], subject to energy and spectrum constraints.

Also to pursue high resolution for a point target, the SF waveform usually has a wide bandwidth, whose frequency changes in the way of off-walk. Its idea is to break down a wide-bandwidth signal into several transmissions, which in turn reduces the instantaneous bandwidth. It is extensively utilized in synthetic aperture radar (SAR) imaging systems by virtue of its excellent anti-interference and high-resolution performance [26,27]. Li et al. [26] proposed an improved imaging algorithm of ultra-high resolution SF spaceborne SAR demonstrating the feasibility of the designed algorithm. Based on terahertz SF, an SAR waveform design method was proposed in [27], which acquired an ultra-high range resolution of 5.2 mm.

The NLFM waveform shapes the power spectrum such that the correlation function exhibits reduced sidelobes. No extra filtering is needed, so that the SNR degradation can be avoided [28]. In [29], an advanced NLFM waveform was developed featuring lower sidelobes and a narrower mainlobe, which acquired a higher SNR of 1.29 dB than LFM. Addressing the issue of excessively high sidelobes in the matched filter response of typical radar signals, a novel NLFM waveform was proposed in [30], with the sidelobe level reduced by at least 10 dB. In [31], a novel method for synthesizing NLFM waveforms has been introduced to achieve low PSLR performance without incurring high computational complexity, which is superior to the robust design algorithms of spherical uncertainty TIR.

Waveform designs for point targets emphasizes spectral characteristics, as the waveform spectrum shapes the autocorrelation function generated by pulse compression, impacting resolution and the suppression of interference from adjacent targets. A larger bandwidth results in a narrower 3dB mainlobe, or higher resolution. Sharp edges or discontinuities in the spectrum lead to sidelobes, which impacts the masking of closely spaced targets. The feature analysis of LFM, SF, and NLFM waveforms are presented in Figure 5. Experimental results demonstrate that the NLFM waveform shows significant sidelobe suppression than LFM and SF waveforms, which highlights the potential interference-resistant capability in radar applications.

#### 3.1.2. Extended Target

As mentioned before, the scattering behavior of extended targets can be characterized by TIR. In waveform optimization process, some papers have utilized a deterministic TIR model, while others have treated it as a random process.

The pioneering work of Bell was the beginning of optimizing radar waveforms for extended targets [5], in which both deterministic and random TIRs were considered to maximize the output SINR and MI, respectively. In [32], an iterative strategy based on maximizing SNR was developed to improve the detection of a deterministic TIR within a signal-dependent interference environment. The work in [32] has been further applied for target identification in [33,34], and such work has been generalized to MIMO radars [35]. However, the techniques in [32,35] were not guaranteed to converge to an optimal solution. To further tackle the problems, the joint design for a transmit waveform and receive filter was introduced in [36], aimed at improving the SINR for a known TIR. Still considering deterministic TIR, the authors studied the optimization of the fully polarized transmitting waveform and receiver TIR to enhance target detection and recognition [37]. The extended target was also modeled as an exact TIR in [38], with optimization procedures for the transmit sequence implemented to improve the performance of target identification and classification.

Unfortunately, the exact knowledge of TIR is required, which is far from practical, since the TIR is highly sensitive to the line of sight [39]. To overcome the problem, TIR is modeled as an uncertainty set or a random vector. Assuming the TIR as partially known, a joint approach was proposed in [40] to optimize both the transmit waveform and receive filter, with the goal of maximizing the SINR for extended targets. The work in [41] still considered the joint design problem presented in [40], incorporating a low PAPR constraint. Hao et al. [42] further applied the waveform design method in [41] to cognitive radar systems, with the goal of estimating the TIR of extended targets. As an extension, when the extended target was moving, a significantly more complicated case was considered in [43], assuming that the TIR was a random process. In [44], a new transmitting waveform with probabilistic robust detection was proposed, but it is only suitable for the cases without clutter. Subsequently, Xu et al. [45] generalized the study in [44] to the signal dependent interference environment. Still assuming unknown knowledge of TIR, the research in [46] established robust design methods to enhance transceiver performance for extended targets, emphasizing the worst-case SINR as the criterion. Besides, an improved adaptive waveform optimization method was proposed in [47] to benefit target tracking and detection.

As we can see, the waveforms for point targets are typically traditional frequency-modulated signals that remain unchanged across different targets, clutter, or noise. Differing from that, waveforms based on extended targets usually do not have explicit expressions but are numerical solutions, which may be varied with different observation scenes. For point targets, waveform design is primarily concerned with optimizing the SNR, resolution, and sidelobe levels. In contrast, for extended targets, greater emphasis is placed on maximizing the SCR and enhancing the scattering characteristics of the target.

### 3.2. Waveform Designs with Different Optimization Criteria

#### 3.2.1. Signal-Power-Based Scenario

There are many works focusing on how to optimize the SNR or SINR, which is an efficient way to improve the performance of target detection. Addressing the notion of a closed-loop intelligent radar in the earlier work of [38], a specific cognitive radar platform was introduced in [48], where SNR and MI were used to adaptively modify transmit waveforms for recognition of known targets. In [49], the waveform design problem was formulated by maximizing the output SINR, subject to energy, similarity and spectral constraints, which was formulated as(19)$$\begin{array}{c} \max_{x}\;x^{H}\mathrm{Rx}\\ s.t.\left\{\begin{array}{c} x^{H}x=1\\ {∥x-x_{0}∥}^{2}\leq \varepsilon \\ x^{H}R_{I}x\leq E_{I} \end{array}\right. \end{array}$$
where R is the inverse of the covariance matrix of the signal-independent interference. Subsequently, an efficient algorithm framework employing feasible point pursuit and successive convex approximation was proposed, demonstrating superior spectral compatibility compared to benchmark algorithms. By maximizing the SINR, the authors [50] considered synthesizing the transmit waveform based on the interference covariance, as shown in Figure 6. Experimental results demonstrated that pulse-compressed output is much better behaved and retains most of the range sidelobe characteristic of the LFM waveform.

Still with the maximize SINR criterion, an adaptive waveform design for extended target detection in cognitive radar frameworks was addressed under sea clutter [51]. In [52], the waveform design together with power allocation was studied for colocated MIMO radar based on compressive sensing, and an iterative algorithm was developed to design the waveforms directly. Additionally, efforts have been made in [36,53,54] to jointly optimize transmit waveforms and receive filters to achieve maximum SNR or SINR. More precisely, a cyclic optimization of the waveform and the receive filter was proposed in [36], which guaranteed nondecreasing SINR performance with each additional iteration. To improve the output SINR, the work in [53] proposed a joint design of transmit and receive weights for colocated MIMO radar.

#### 3.2.2. Information-Theory-Based Scenario

As early as in 1993, the MI between targets and echoes was first introduced as a criterion for optimizing transmit waveforms, with maximum MI under pure noise environment [5]. In [36], a scheme on the basis of “water-filling” method to design transmit waveform was researched by utilizing maximum MI criterion under MIMO radar. Yang and Blum [6] utilized the MI between echo and scattering signals as the criterion for designing MIMO radar waveforms, yielding outcomes comparable to the minimum mean-square error approach. The work in [7] extended the findings of [6] to scenarios with colored noise, designing a novel MI-based transmit waveform for MIMO radar, and a comparison between MI and relative entropy criteria was presented. Goodman [38] adopted Bell’s method for cognitive radar by incorporating a feedback loop in the transceiver. Leshem [8] expanded upon the maximum MI waveform design approach to enhance tracking of multiple targets in noisy environments. In [9], an approach for optimizing the transmit weight sequence and receive filter was proposed to enhance target information acquisition in SAR, with the MI serving as the objective function, and the optimization problem was expressed as(20)$$P_{x,w}\left\{\begin{array}{c} \max_{x,w}\;f\left(x,w\right)\\ s.t.\left\{\begin{array}{c} \|x⊙c{\|}^{2}=E_{x}\\ \|x⊙c-c{\|}^{2}\leq \varepsilon_{1}\\ \|w{\|}^{2}=E_{w}\\ \|w-r{\|}^{2}\leq \varepsilon_{2} \end{array}\right. \end{array}\right.$$
where $\varepsilon_{1}$ and $\varepsilon_{2}$ represent the similarity parameters, $E_{x}$ and $E_{w}$ are energy of the waveform and the receive filter w, respectively. As a result, the imaging effect of the designed waveform was compared with other classical waveforms, as shown in Figure 7. Experimental results demonstrated that the SAR images with the designed waveform yielded the best contrast between target and background. In [10], the optimal radar waveform was derived for detecting a spectrum-spread target in the presence of clutter by maximizing MI and the KLD. The study in [11] proposed a cooperative optimization scheme based on MIMO radar-communication spectral sharing system, with the MI still being maximized. For airborne MIMO radar waveform design, Sun [12] employed MI as the optimization metric and enhance ground moving target detection performance by combining MM and ADMM techniques.

Focusing on enhancing the detection performance of radar systems for multi-tasks, some papers have carried out waveform design under relative entropy criterion. In [43], the optimal radar waveform design was considered by maximizing the KLD, to improve the detection performance for a moving doubly spread target. The work in [10] addressed the issue of optimal radar waveform design for detecting a spectrum-spread target in the presence of clutter, deriving the solution by maximizing both MI and KLD. Zhu et al. [55] analyzed the trade-offs among the metrics of KLD, MI, and SNR that were widely utilized in waveform design. Using the relative entropy criterion, a novel waveform design approach was proposed in [56] to enhance the detection performance of MIMO radar, and an iterative method based on the MM technique was devised to tackle the non-convex design problem. In [57], the joint radar and communications with orthogonal frequency division multiplexing waveform transmission was proposed where the performance of radar detection was measured by KLD. In [58], the KLD criterion was further employed in adaptive waveform design, focusing on maximizing the practical resolution of a specific radar system.

#### 3.2.3. Ambiguity-Function-Based Scenario

As one of the main performance metrics, AF regulates both range and Doppler resolutions. An ideal AF for radar transmitted waveforms should exhibit a thumbtack-like shape, featuring a peak at the target of interest while minimizing interference energy [59]. Unfortunately, achieving an ideal AF is challenging with limited information in traditional radar waveform design problems, as the focus is mainly on reducing self-clutter by relying on signal characteristics independent of the surrounding environment [60]. The work in [61] considered a waveform design method that combined the AF and energy spectral density to enhance the detection probability of cognitive radar targets. To shape the AF, Cui et al. [62] proposed an accelerated iterative sequential optimization algorithm by minimizing the average value of the weighted integrated sidelobe level. Furthermore, the work in [63] studied the problem of multistatic radar by transmitting an orthogonal LFM waveform, and the target resolution capabilities of the waveform was analyzed based on AF. Xu et al. [64] focused on the MI-based waveform design problem with high-resolution constraint, and the AF between the proposed waveform and the linear frequency modulated waveform (LFMW) was evaluated, as shown in Figure 8. As we can see, the AF of the designed waveform approximates a pin shape with a single peak at zero delay and Doppler, as shown in Figure 8a,b, indicating nearly ideal range-Doppler ambiguity properties. In contrast, the AF of LFMW in Figure 8c,d resembles a blade, exhibiting range-Doppler coupling due to its oblique triangular shape.

More generally, several methods have been proposed to shape the AF in terms of the PSLR or ISLR as metrics, which are also important indexes to evaluate the sidelobe suppression performance in radar waveform design [65]. To reduce PSLR, many efforts have been made in recent years [66,67,68,69,70]. A PSLR minimization problem under the spectral constraint was proposed in [66], providing a waveform design method based on alternating minimization with more flexibility in spectral adjustment. In [67], a minimum weighted peak sidelobe level design problem for waveform synthesis was constructed, subject to the spectral and PAPR constraints. Still pursuing the lower PSLR, an iterative majorizer algorithm was proposed in [68] to optimize waveform and mismatched filter.

The above works focused on the PSLR reduction without taking into account the influence of interference. The interference cross-correlation peak level is also a crucial issue for target detection in interference environment. In [69], an approach to jointly design the transmit waveform and receive filter was considered to improve weak target detection and interference suppression. In [70], the worst-case ISLR formed as the objective function was considered in the robust emission beampattern synthesis of MIMO radar, ensuring improved worst-case performance compared to other counterparts. In [71], the transmit waveform and receive filter were jointly designed to maximize target detection in colocated MIMO radar systems, achieving high SINR and low output ISL. Still pursuing low ISL, Raei et al. [72] subsequently extended the work in [71] to cognitive colocated MIMO radar systems. Meanwhile, the joint design of waveform and filter developed in [73] for sidelobe suppression achieved a lower ISLR with minimal loss in processing gain, outperforming methods that focus solely on waveform optimization. The study in [74] proposed a strategy to suppress deceptive interference by jointly designing the transmit waveform and the receive filter, where the optimization problem is written as(21)$$\begin{array}{c} \min_{x,w}\;\lambda_{1}\mathrm{ISLR}\left(x,w\right)+\lambda_{2}\mathrm{ISLR}\left(x_{J},w\right)\\ s.t.\left\{\begin{array}{c} x^{H}R_{\mathrm{wj}0}x\leq \varepsilon \;\mathrm{or}\;w^{H}R_{\mathrm{xj}0}w\leq \varepsilon \\ |x(k)|=1\\ w^{H}x=N\\ \frac{N^{2}}{\|w{\|}^{2}}\geq \eta \end{array}\right. \end{array}$$
with $R_{\mathrm{wj}0}=\frac{1}{P}\underset{p=1}{\sum^{P}}\;J_{p}^{H}ww^{H}J_{p},R_{\mathrm{xj}0}=\frac{1}{P}\underset{p=1}{\sum^{P}}\;J_{p}xx^{H}J_{p}^{H}$, and the jamming shifting matrix corresponding to the pth error is defined as $J_{p}$. The resulting non-convex problem was solved by a novel decoupled alternating direction penalty method, which improved the ability to resist deceptive interferences. The work in [75] provided a comparison of the correlation levels of three sequences, as shown in Figure 9. Under the ISLR metric, the cyclic algorithm-new (CAN) sequence initialized by the Golomb sequence CAN(G) offered lower ISLR than either the m-sequence or the random-phase sequence. Additionally, the author still compared the ISLR values of CAN sequences and P4 subject to the constraint of PAPR≤ρ, as shown in Figure 10.

### 3.3. Waveform Designs with Different Constraints

#### 3.3.1. Energy Constraint

Since the power of radar transmitter is limited, energy constraint is always imposed on the waveform [76,77,78,79,80,81,82]. Inspired by the work in [76], a robust waveform design method was considered to enhance multi-target detection performance in cognitive MIMO radar, subject to energy constraint [77]. In [78], the issue of code design for space-time adaptive radar was addressed involving energy and similarity constraints. The work in [78] was further investigated in [79] under the PAPR and energy constraints. With the same constraints considered in [80], an efficient algorithm utilizing the MM technique was employed to address large-scale beampattern design problems in MIMO radar. Besides, the authors dealt with the joint design of transmit waveform and receive filter bank for airborne MIMO radar, still considering the energy constraints of the transmit waveform [81]. More recently, a novel waveform design method has been proposed to tackle the issue of weak target masking in congested spectral environments, which simultaneously shaped both the AF and the waveform spectrum incorporating the energy and PAPR constraints [82].

#### 3.3.2. Constant Modulus Constraint

Due to the limitations of the hardware components used for signal generation, a constant modulus or PAPR constraints is often imposed on the transmit waveform. Although such constraints are very important in the field of waveform design, they will make the optimization problem NP-hard [79]. In [83], constant modulus waveforms and receive filters were jointly designed, focusing on enhancing the detection performance of extended targets. In [84], a constant modulus transmit waveform was designed for multi-target detection in MIMO radar systems. Still with the performance of the system [84], Wang et al. [85] designed constant modulus probing waveforms with low correlation sidelobes. Soon after, they extended the previous work of [85] and a consensus-ADMM approach was proposed to design constant modulus probing waveforms [86]. It offered lower computational complexity and sped up the convergence, benefiting large-scale MIMO radar systems. The work in [44] considered the PAPR constraint for extended targets, where a novel waveform was proposed based on probabilistically robust properties metric. Still imposing the PAPR constraint, a customized algorithm was introduced in [87] to overcome the limitations of MIMO transmit beampattern synthesis that applied only to narrowband signals, thereby enhancing its suitability for wideband MIMO systems. In [88], a two-step strategy was introduced that contrasted with the one-step method presented in [52], which improved the computational efficiency and imposed constant modulus constraint on the waveform. Besides, Shi et al. [89] addressed the waveform design to facilitate spectrum co-existence between MIMO radar and communication systems, subject to the ISLR and PAPR constraint. Subsequently, an iterative algorithm based on ADMM framework was proposed, which effectively decomposed the original problem into manageable subproblems.

#### 3.3.3. Spectral Constraint

A spectral constraint is $x^{H}R_{I}x\leq E_{I}$, where $R_{I}$ is used to calculate the energy of the waveform within a specific frequency band, and $E_{I}$ is the maximum allowed interference energy tolerated by other radiators. The degree of the spectral constraint can be controlled by adjusting the value of $E_{I}$.

Over the past decades, spectrum resources have become a focal point, driven by pressing challenges posed by a crowded radio frequency spectrum, thus the aspect of waveform design has been put on spectral constraint [90,91,92,93,94,95,96,97]. In [90], an efficient approach to synthesize radar waveform was considered in spectrally crowded environments, with fixed transmitted waveform energy. Aubry et al. [91] extended the work in [90] by incorporating the modulation of transmitted waveform energy. It offered greater degrees of freedom for the optimization process, enabling the radar system to tackle complex spectral constraints. Different from [90,91], an optimization technique to synthesize waveforms was further proposed in [92], while ensuring spectral coexistence with some overlaid radiators. Without concerning for signal-dependent interference, the work in [94] addressed radar waveform design in spectrally crowded environments, and multiple spectral constraints were taken into account. Subsequently, Yang et al. [95] improved the work of [94], and the joint synthesis of constant envelope transmit waveform and receive filter was considered in signal-dependent interference. The work in [96] addressed the design of one-bit transmit sequences for MIMO radar under spectral constraints, demonstrating improved spatial and spectral properties with lower computational complexity compared to existing methods. More recently, Zhou et al. [97] addressed the robust radar waveform design for extended target detection, and the multiple spectral compatibility constraints were imposed on the waveform. Inspired by [98], we plot the energy spectral density (ESD) of an designed waveform in Figure 11. It can be found that the spectral constraint forces the ESD shape to have a deep notch over the stopbands. The nulls are deeper for smaller $E_{I}$, as less energy is distributed over the frequency band. It indicates that the spectral constraint is effective to achieve spectral compatibility.

#### 3.3.4. Similarity Constraint

Similarity constraints are commonly considered in waveform optimization design to ensure that the designed waveform meets some desired characteristics of the reference waveform, such as high resolution and easy generation. It is usually written as ${∥x-x_{0}∥}_{2}^{2}\leq \varepsilon$, where $x_{0}$ is the reference signal. By appropriately setting the similarity parameter ε, the optimized waveform can retain the characteristics of the reference one.

To guarantee the designed waveform exhibiting advantages such as high range resolution, low peak sidelobe levels, and better AF, the similarity constraint is often incorporated into the optimization problem to improve practical feasibility [99]. In [100], the problem of MIMO radar waveform design was studied under signal-dependent interference and white Gaussian noise, subject to constant modulus and similarity constraints. Still using the similarity constraint, the issue of spectrally compatible waveform design was addressed in [101], outperforming other existing works. The work in [102] considered the issue of spectrally compatible waveform design for a colocated MIMO radar under the PAPR and similarity constraints. Based on [103], the effects of similarity parameters on chirp signal and AF were provided, respectively, as shown in Figure 12 and Figure 13. Simulation results demonstrated that their characteristics gradually diverge from the benchmark as ε increases, confirming the feasibility of similarity constraint in practical applications.

### 3.4. Waveform Designs with Different Optimization Algorithms

#### 3.4.1. ADMM-Based Methods

Based on the ADMM, some representative examples can be found in [86,89,104,105,106,107]. In [104], a double cyclic alternating direction method of multipliers (D-ADMM) algorithm was introduced for colocated MIMO radar systems to address the non-convex beampattern design problem. The approach was further applied in [105], where the joint optimization problem with respect to the covariance matrix and antenna position was solved. In [106], the authors considered the joint design of transmit and receive beamforming in the MIMO radar system, and the ADMM technique was employed to decouple variables maximizing the output SINR. Still with the same framework [106], a similar ADMM-based approach has been further investigated in [107] to obtain an optimal waveform. Shi et al. [89] proposed a novel radar waveform to guarantee the spectrum co-existence between MIMO radar and communication systems, and the ADMM technique was employed to address the resulting NP-hard problem. However, the authors [89] failed to produce a waveform with satisfactory attenuation in the stopbands due to the dominance of the ISLR constraint, which prevented the cost function and PAPR constraint from achieving the desired outcomes simultaneously. Subsequently, an optimization problem was formulated by introducing a constraint on the energy spectral density, which was then solved by the ADMM-based technique [108].

#### 3.4.2. SDR-Based Methods

Based on the SDR algorithm, two sequential optimization methods were introduced in [100] to maximize the output SINR, subject to constant modulus and similarity constraints. However, incorporating the latter in SDR-based algorithms led to a poor output SINR. To this end, the work in [99] proposed two novel algorithms to fill this gap based on the successive QCQP refinement. In [109], a customized algorithm for designing transmit waveforms in MIMO radar was proposed, with the SDR technique was employed to address the resultant non-convex problem. The work in [110] realized the optimal design of interpulse phase-coded signal in the background of clutter and noise by using the SDR method. To optimize the worst-case SINR, a robust joint design approach for transmit waveform and receive filter was introduced in [76], with rank-one relaxation and SDP being taken into account. Still employing the SDP algorithm, the authors [111,112] matched the waveform covariance matrix to the desired beampattern, synthesizing the constant modulus waveforms. The same technique was further extended to the multi-target scenario in [113], resulting in the proposal of an adaptive waveform for cognitive MIMO radar. In [114], the SDR technique was utilized to tackle the NP-hard challenges associated with the joint design of transmit and receive beamforming weights.

#### 3.4.3. MM-Based Methods

In [17], the problem was formulated under the MM framework to jointly design the transmit waveform and receive filter, with the purpose of maximizing the output SINR in an MIMO radar system. The same technique was also employed in [71] for joint design of MIMO radar, offering the trade-off between the output SINR and ISLR under the PAPR constraint. A new optimization model was formulated in [115] to enhance the detection performance of multi-static radar by employing several information-theoretic criteria, and the MM method was employed to tackle the optimization problems. Besides, Zhao et al. [116] proposed a one-step scheme to directly address the MIMO transmit beampattern matching issue, achieving faster convergence compared to D-ADMM [104,105]. The work in [117] emphasized the cross AF synthesizing problem by jointly optimizing the transmit waveform and receive filter, subject to a PAPR constraint, addressing the associated challenges with an efficient MM algorithm. Still employing the MM algorithm, three subproblems were formulated in [118] to optimize transmit waveform x, range filter m, and azimuth filter w. The objective function and surrogate function were used to verify its effectiveness, as shown in Figure 14. As we can see, the method was feasible in tackling the three subproblems, and it converged rapidly, further confirming the MM algorithm was effective.

In waveform design, these algorithms provide a trade-off between metrics and operational constraints. ADMM algorithm has been widely utilized for solving large-scale convex and non-convex problems, demonstrating superior performance in handling sparsity constraints in waveform design. SDR technique has shown efficacy in approximating non-convex problems to convex formulations, enabling solutions to complex radar system requirements. Moreover, MM algorithm has been effective in addressing nonlinear optimization problems, particularly in applications requiring iterative refinement for sidelobe suppression or time-frequency optimization. To evaluate the complexity of the ADMM, SDR, and MM methods, these algorithms are used to solve the optimization problem provided in [119] to evaluate time consuming, as shown in Figure 15. It can be found that the ADMM method proved more efficient as the number of waveform samples increased, making it suitable for large-scale data. The SDR technique had the highest time consuming, with lower efficiency during processing, while the MM algorithm performed intermediate to the other methods.

## 4. Research Challenges of Radar Waveform Design

Current research contributions encompass various optimization criteria, constraints, target models, and optimization methods. Although waveform design problem has been widely studied in recent decades, it is still far from being fully solved. A significant challenge lies in multimodal sensing collaborative detection, where integrating optical, radar, and lidar data holds immense potential for enhancing detection accuracy, target recognition, and environmental monitoring. Achieving seamless harmonization of data from these diverse sources requires innovative waveform optimization techniques to fully exploit their complementary strengths. Another critical issue is the design of waveforms for multi-tasking, which focuses on how to simultaneously perform multiple functions such as target detection, tracking, and imaging within the same system. Additionally, modern radars often capture poor target information, making the optimization of waveforms based on sparse signal recovery increasingly relevant for future applications. Finally, the development of machine learning and artificial intelligence technologies also highlights the importance of integrating environmental perception into pulse radar waveform design.

As mentioned above, the difficulties currently faced in waveform design span multiple levels, including technological, environmental, and application aspects. To this end, this paper identifies the following key development trends in the next few decades.

### 4.1. Radar Waveform Optimization for Multimodal Sensing Collaborative Detection

Advancing land monitoring through the synergistic harmonization of optical, radar, and lidar satellite technologies requires the integration of complementary data from these diverse sources. Multimodal sensing collaborative detection leverages the unique strengths of optical sensors for high-resolution imagery, radar systems for all-weather capabilities, and lidar for precise elevation and structural information. This integrated approach significantly enhances detection accuracy, target identification, and environmental tracking in complex monitoring scenarios, particularly in regions with high interference or clutter where single-sensor systems are insufficient.

However, current waveform optimization techniques largely focus on improving single-sensor detection capabilities [9,10,12], neglecting the intricacies of multimodal sensing harmonization. They are unable to effectively support the seamless integration of diverse sensor inputs, limiting their potential in collaborative scenarios. Thus, future research should focus on the collaborative design of radar waveforms tailored for integration with optical and lidar technologies. By leveraging the complementary advantages of multi-waveform and multimodal sensing integration, the goal is to enhance feature extraction, environmental monitoring, and target recognition. Efficient data fusion methods to harmonize heterogeneous sensor outputs will be critical for realizing the full potential of this synergistic approach, paving the way for advanced capabilities in land monitoring and environmental analysis.

### 4.2. Radar Waveform Optimization for Joint Multi-Tasking

Although the existing waveform design methods typically focus on enhancing detection capability and resolution for a typical target, modern radars require multitasking capabilities to acquire target information. Thus, the waveform optimization designs will integrate multiple tasks and evolve towards various optimization criteria, simultaneously considering factors such as AF characteristics [61,62,63], similarity constraints [54,99,100,101,102], detection probability [43,44,45,46,56,77,113] and parameter estimation performance. Achieving the execution of multiple tasks simultaneously imposes higher demands on waveform design.

To achieve this goal, more advanced optimization algorithms need to be employed to design transmission waveforms which simultaneously meet the requirements of multiple tasks. For example, in scenarios involving both target detection and communication, radar waveforms need to provide high-resolution detection capabilities while maintaining a stable communication channel. In complex and dynamic environments, different tasks have varying priorities and real-time requirements, necessitating that radar systems dynamically adjust waveform parameters based on the importance of each task and its real-time demands.

### 4.3. Radar Waveform Optimization Based on Sparse Signal Recovery

With the advancement of computational capabilities and the growing maturity of compressed sensing theory, waveform optimization design based on sparse signal recovery has demonstrated significant potential. By designing sparse radar waveforms and integrating compressed sensing techniques, radar systems can achieve efficient signal reconstruction and target identification, even at reduced sampling rates.

In multi-target detection and tracking scenarios, traditional waveforms often need to handle a large number of signal sources simultaneously, leading to inefficiencies. Fortunately, waveform optimization based on sparse signal recovery can effectively separate and extract the sparse features of different targets, enabling precise detection and tracking of multiple targets. Nonetheless, research on sparse signal recovery in specific areas still has limitations. Although the designed waveform in [52,88] is applicable to small-scale detection and tracking tasks, it is not easy for imaging radar due to the limitations of transmission bandwidth. Given the above problems, waveform design based on sparse signal recovery is expected to be a key focus of future research.

### 4.4. Radar Waveform Optimization with Intelligent Perception for Environmental Adaptation

In the current complex and varied environment, radar waveform design needs to incorporate intelligent perception algorithm to effectively cope with environmental changes, but it has not achieved the expected effect.

The application of machine learning and artificial intelligence technologies in waveform design will drive radar waveform designs, necessitating human adaptation to the new and unique challenges posed. By training deep learning models, radar systems can learn from historical data and predict dynamic environmental changes, allowing for the dynamic adjustment of waveform parameters to adapt to real-time environmental characteristics. Although there have been many works designing intelligent sensing algorithms [38,48], there are still some limitations. First, the effectiveness of machine learning and artificial intelligence algorithms heavily depends on the availability of large amounts of high-quality data, which imposes higher demands on radar waveform performance. Additionally, the process of sensing, analyzing and adjusting waveforms in real time introduces delays, which can adversely affect system performance in rapidly changing scenarios, and will require further research in the future.

To enable effective collaboration of radar waveforms across multimodal sensors, a potential solution involves adopting a joint optimization approach that accounts for the response characteristics of different sensors as well as interference suppression. For example, a multi-sensor joint optimization algorithm optimizes radar waveform spectral and temporal properties, facilitating complementary detection when integrated with other sensors. In joint multitasking radar waveform optimization, a dynamic task prioritization mechanism allows flexible adjustment of waveform characteristics based on specific task demands. To address sparse signal recovery challenges, compressive sensing theory integrates the signal compression and recovery process into the waveform design, reducing computational complexity and enhancing system robustness. In intelligent environmental sensing, adaptive algorithms incorporated into radar waveform design enable real-time adjustments based on environmental changes, target characteristics, and interference conditions.

## 5. Conclusions

Radar systems rely on active sensing with customized waveforms to enhance target detection and signature extraction, making waveform design a critical factor in system performance. Transmitted waveforms are tailored to specific target characteristics, whether they appear as point-like or extended targets. The design process is inherently challenging due to the high-dimensional and non-convex nature of the optimization problems involved, compounded by rigorous constraints such as energy limits, constant modulus, and sidelobes suppression. To tackle the various waveform optimization problems, it usually to customize methods based on traditional ADMM, SDR, and MM algorithms to tackle specific optimization problems, achieving a balance between time consumption and objective function values. Although there have been fertile works, challenges persist in achieving effective multimodal sensing collaborative detection, joint multi-tasking, sparse signal recovery, and intelligent perception for diverse environments. Future research may focus on techniques such as cross-modal information integration, multi-objective optimization, compressive sensing, and adaptive algorithms in waveform design.

## Figures and Tables

**Figure 1 sensors-25-01203-f001:**
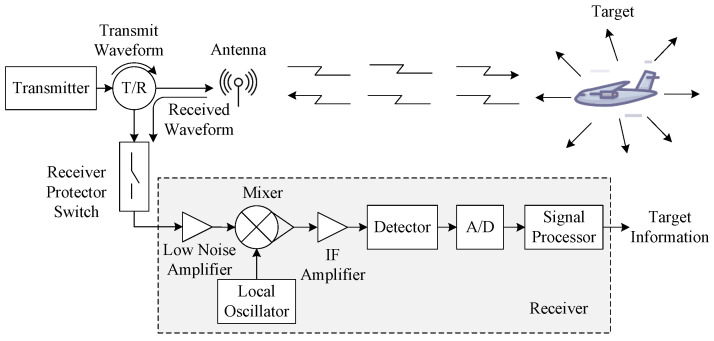
The principle of radar transmit-receive operation.

**Figure 2 sensors-25-01203-f002:**
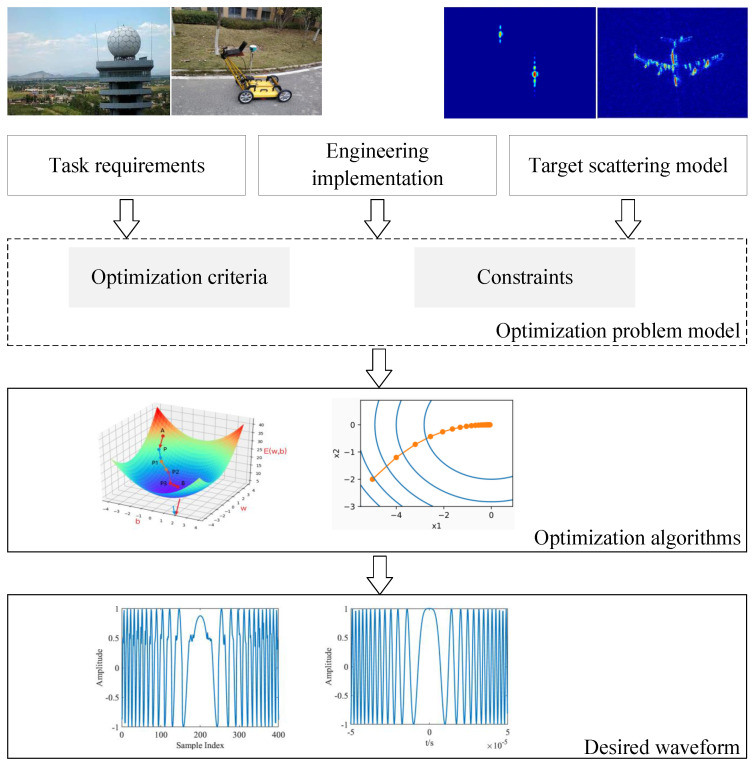
Schematic diagram of radar waveform design.

**Figure 3 sensors-25-01203-f003:**
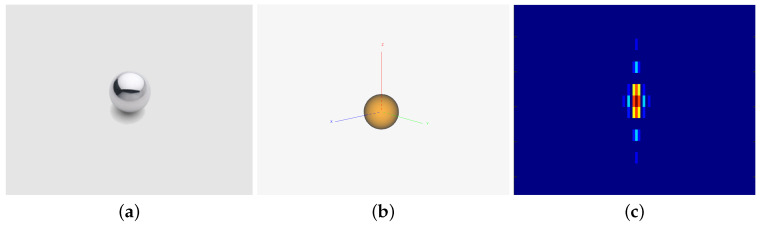
Point-like target: (**a**) optical image of a metal sphere, (**b**) the model of an ideal metal sphere, and (**c**) its ISAR image.

**Figure 4 sensors-25-01203-f004:**
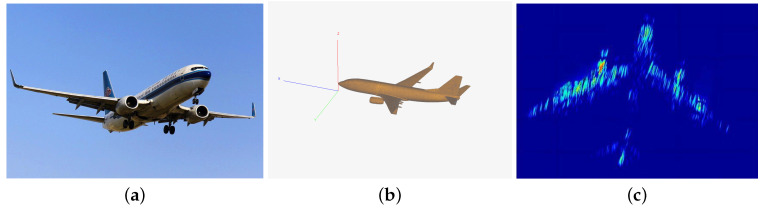
Extended targets: (**a**) optical image of a Boeing 737 aircraft, (**b**) the model of an ideal Boeing 737 aircraft, and (**c**) its ISAR image.

**Figure 5 sensors-25-01203-f005:**
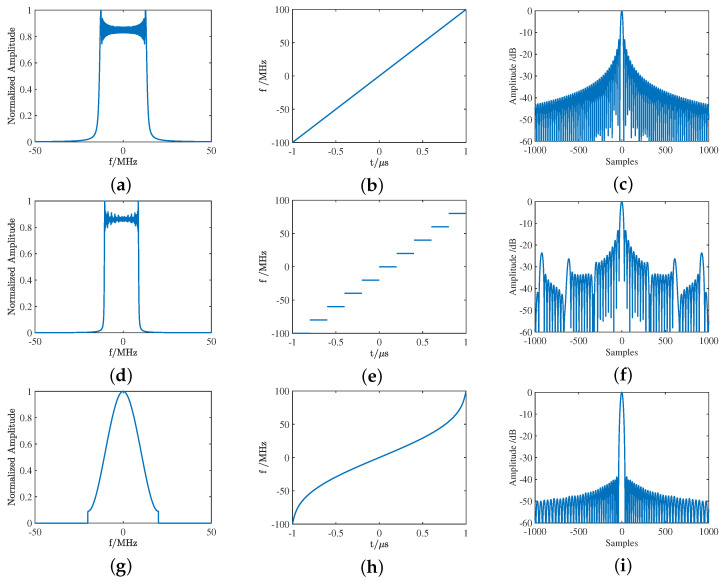
Feature analysis with different signals: (**a**) spectrum of the LFM waveform, (**b**) time-frequency characteristics of the LFM waveform, (**c**) correlation function of the LFM waveform, (**d**) spectrum of the SF waveform, (**e**) time-frequency characteristics of the SF waveform, (**f**) correlation function of the SF waveform, (**g**) spectrum of the NLFM waveform, (**h**) time-frequency characteristics of the NLFM waveform, and (**i**) correlation function of the NLFM waveform.

**Figure 6 sensors-25-01203-f006:**
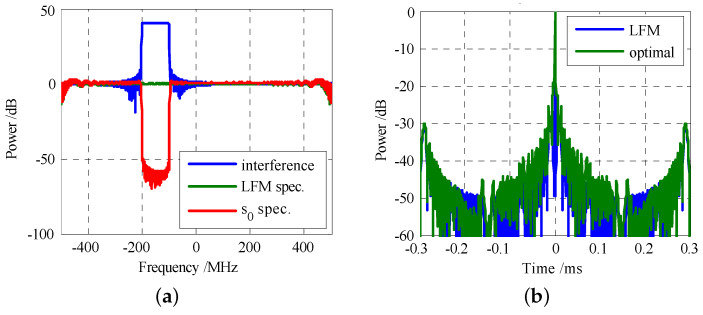
The spectra of the interference and the optimal waveform: (**a**) waveform spectra, (**b**) corresponding time series.

**Figure 7 sensors-25-01203-f007:**
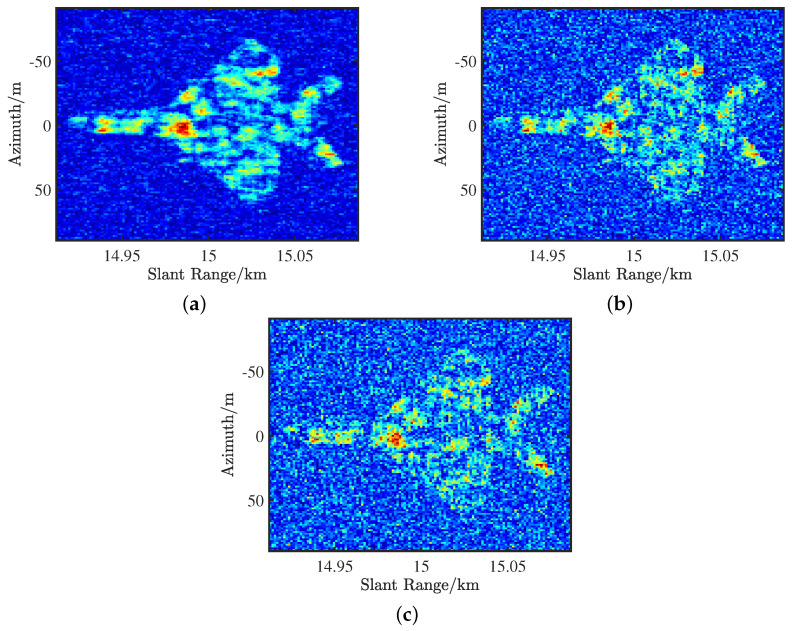
SAR images with different waveforms: (**a**) the weighted LFM signal, (**b**) the classical LFM signal, and (**c**) the classical P4 sequence.

**Figure 8 sensors-25-01203-f008:**
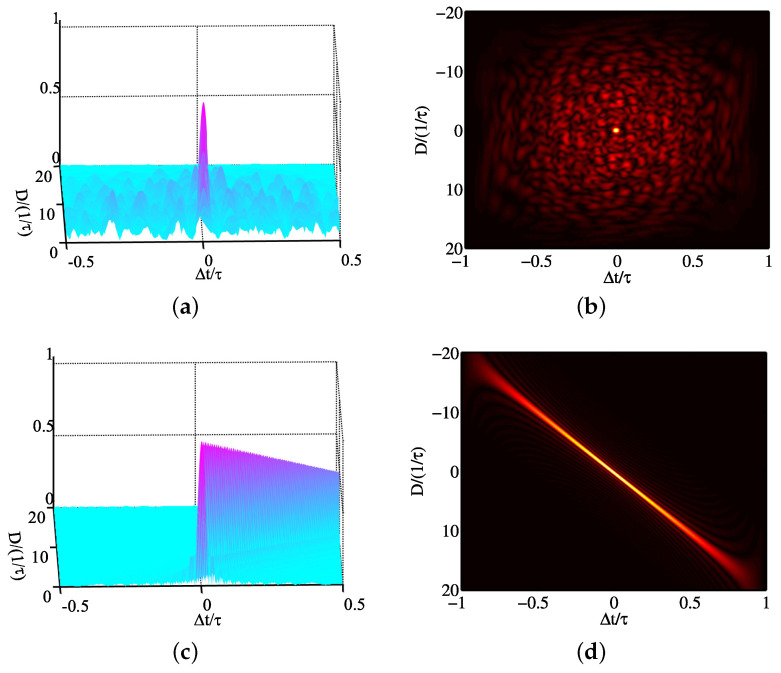
Partial 3-D and 2-D AF of the proposed waveform and LFMW: (**a**) 3-D AF of the designed waveform, (**b**) 2-D AF of the designed waveform, (**c**) 3-D AF of LFMW, and (**d**) 2-D AF of LFMW. Δt, *D*, and τ represent time delay, Doppler frequency shift, and duration, respectively.

**Figure 9 sensors-25-01203-f009:**
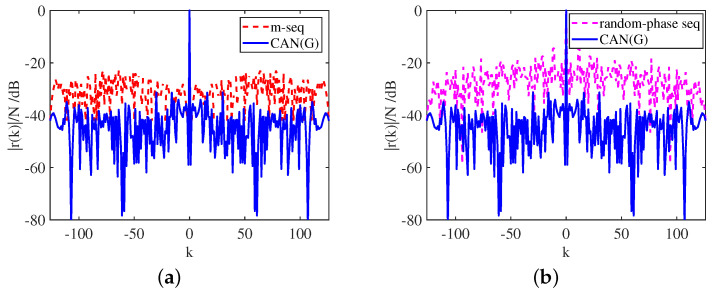
Correlation levels of the CAN(G), m- and random-phase sequences of length *N* = 127, designed under the ISLR metric: (**a**) the CAN(G) and m-sequences, (**b**) the CAN(G) and random-phase sequences.

**Figure 10 sensors-25-01203-f010:**
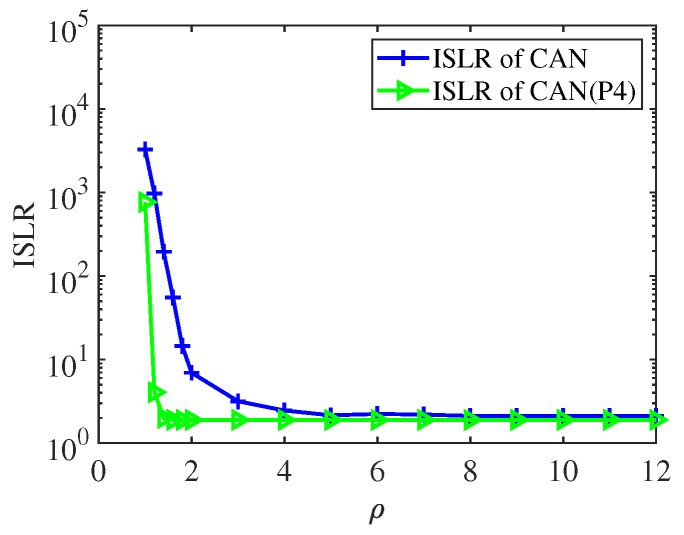
The ISLR of the CAN sequences (with length *N* = 256 and initialized either randomly or by P4) versus ρ.

**Figure 11 sensors-25-01203-f011:**
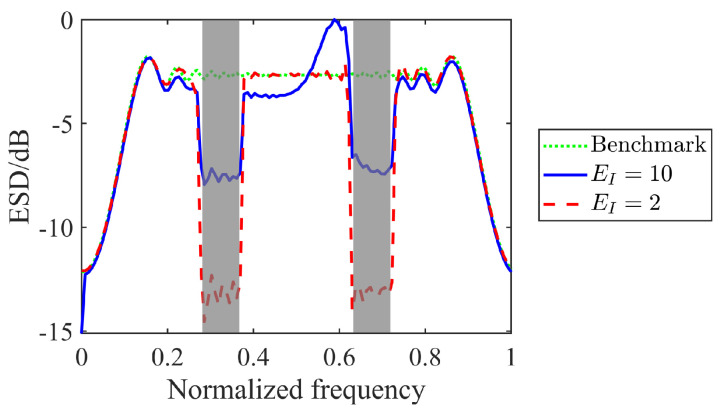
ESDs of designed waveforms versus varied $E_{I}$, where the stopbands are shaded in light gray.

**Figure 12 sensors-25-01203-f012:**
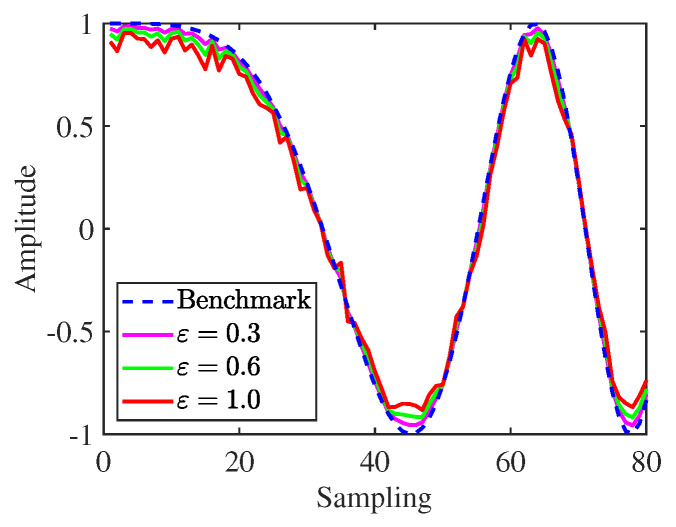
Similarity constraints under different parameters.

**Figure 13 sensors-25-01203-f013:**
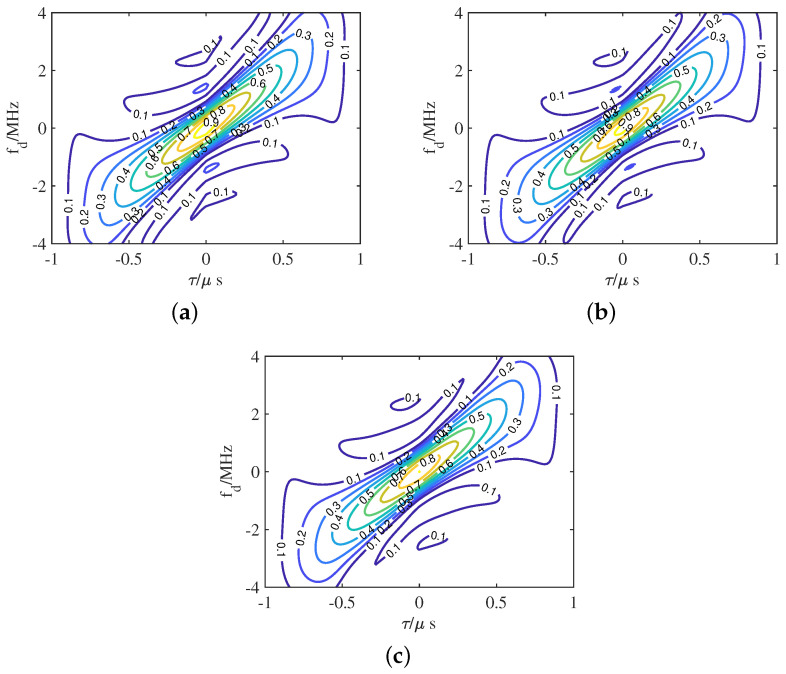
Ambiguity function with different similarity parameters: (**a**) ε = 0.3, (**b**) ε = 0.6, and (**c**) ε = 1.0.

**Figure 14 sensors-25-01203-f014:**
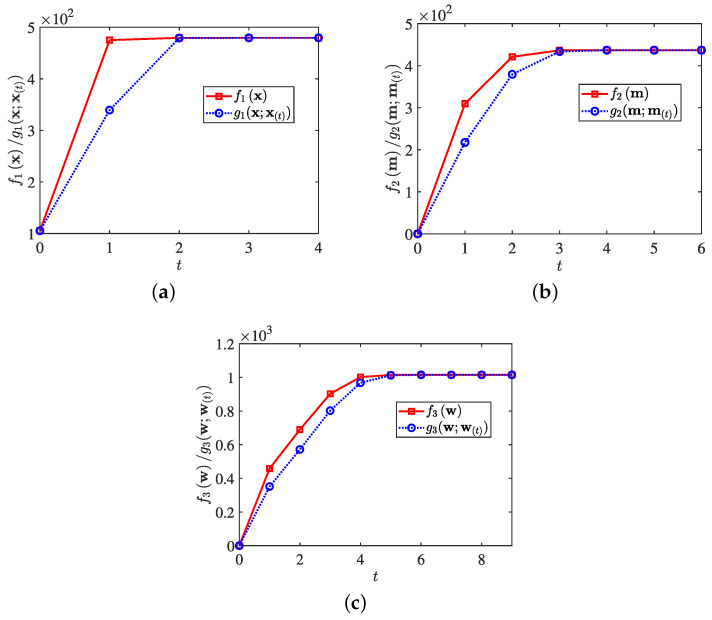
Objective function and surrogate function of subproblem versus iteration step: (**a**) $f_{1}\left(x\right)$ and $g_{1}\left(x;x_{t}\right)$, (**b**) $f_{2}\left(m\right)$ and $g_{2}\left(m;m_{\left(t\right)}\right)$, and (**c**) $f_{3}\left(w\right)$ and $g_{3}\left(w;w_{\left(t\right)}\right)$.

**Figure 15 sensors-25-01203-f015:**
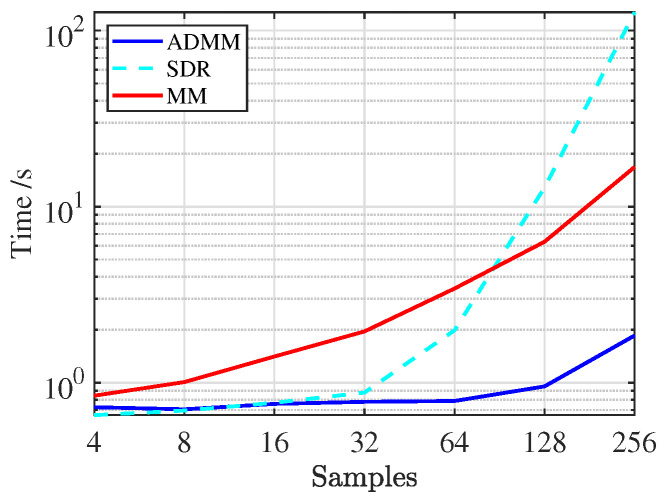
Time consuming of ADMM, SDR, and MM algorithms.

**Table 1 sensors-25-01203-t001:** Constraints of radar waveform design.

Constraint Condition	Expression Formula
Energy constraint	$x^{H}x\leq E,$ $\int_{-W/2}^{W/2}|X\left(f\right){|}^{2}df\leq E$
Constant modulus constraint	‖x(k)‖=1,k=1,2,⋯
Low PAPR constraint	$\begin{array}{cc} \mathrm{PAPR}\left(x\right) & =\max_{k}\;|x_{k}{|}^{2}/\left(\frac{1}{N}\sum_{k=1}^{N}\;|x_{k}{|}^{2}\right)\\ & =\max_{k}\;|x_{k}{|}^{2}/\left(E/N\right)\leq \mu \end{array}$
Spectral constraint	$x^{H}R_{I}x\leq E_{I},R_{I}=\underset{q=1}{\sum^{Q}}\;\gamma_{q}\Omega_{q}$
Similarity constraint	$\|x-x_{0}{\|}^{2}\leq \varepsilon$

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
