# Peer review of "Constrained Pulse Radar Waveform Design Based on Optimization Theory"

_sensors, 2025, doi:10.3390/s25041203_

Round 1
Reviewer 1 Report
Comments and Suggestions for Authors
Please see the attachment.

Reviewer 2 Report
Comments and Suggestions for Authors
1 The reviewed work is devoted to a review of methods for optimizing radar operation based on improving the signal shape, taking into account possible limitations. The topic of the article is relevant, since radars are currently actively used in various areas of practical activity, including remote sensing of the Earth, geological research, in tasks related to security, climate research and many others.
2. Unfortunately, the article is overloaded with references (121 references), and the authors of the review provide too brief a description of each publication, from which it is very difficult to draw the right conclusions without reading the publication. It would seem that the authors should have highlighted the main publications and drawn the reader's attention to them in Section 5 (Conclusion).
3. It would be useful to expand the introduction by describing the subject of the study in more detail and answering the question: what successes have already been achieved and what problems need to be solved in the future.
Notes:
4. The authors present formulas that are apparently taken from the cited works, but since they are not described in detail and specific examples of their application are not given, these formulas do not clarify much.
5 The captions to Fig. 8 should be explained. The discussion of the figure should be more detailed.
6. It is worth explaining what EI in Fig. 11 is, which is "used to limit energy", and how it is used.
7. Fig. 12 and Fig. 13 are difficult to compare, since the parameter ε in Fig. 12 takes the values 0.3, 0.6, 1, and in Fig. 13 - the values 0, 0.3, 0.8. What are the values on the level lines? (Fig. 13)
Reviewer 3 Report
Comments and Suggestions for Authors
This work examines the design of constrained radar waveforms within the framework of optimization theory. The content is generally comprehensive, yet several enhancements are proposed for consideration:
1、In its capacity as an introductory section for a review article, the content could be augmented with more detailed real-world application scenarios to enhance reader engagement. Furthermore, emphasizing recent research breakthroughs would underscore the state-of-the-art nature of the field.
2、The paper delves into several pivotal technical challenges; however, the inclusion of novel solutions to address these challenges could significantly enrich the discussion.
3、It is advisable to append a section at the conclusion of the introduction that elucidates the specific contributions of this paper.
4、Although the paper presents results pertaining to diverse targets and optimization theories, it lacks a comparative analysis of various methodologies. Specifically, comparing the outcomes of different optimization theories applied to identical target scenarios is absent.Given the detailed introduction of multiple optimization theoretical methods, it may be prudent to undertake a comparative analysis and summarize the distinctions in computational complexity among these algorithms.
